# Regulation of Mindfulness-Based Music Listening on Negative Emotions Related to COVID-19: An ERP Study

**DOI:** 10.3390/ijerph18137063

**Published:** 2021-07-01

**Authors:** Xiaolin Liu, Yong Liu, Huijuan Shi, Ling Li, Maoping Zheng

**Affiliations:** 1Key Laboratory of Cognition and Personality (Ministry of Education), Southwest University, Chongqing 400715, China; Liumusicpsy@163.com (X.L.); liuy0768@swu.edu.cn (Y.L.); 2School of Psychology, Southwest University, Chongqing 400715, China; 3Institute of Chinese Music Aesthetic Psychology and Basic Theory of Music Performance, Chongqing Institute of Foreign Studies, Chongqing 401120, China; shishi1984219@sina.com (H.S.); llling19954@126.com (L.L.); 4School of Music, Southwest University, Beibei, Chongqing 400715, China

**Keywords:** mindfulness meditation, music listening, emotion regulation, cognitive control, ERPs

## Abstract

The current study aimed to explore the behavioral and neural correlates of mindfulness-based music listening regulation of induced negative emotions related to COVID-19 using the face–word Stroop task. Eighty-five young adults visited the laboratory and were randomly assigned to three groups: a calm music group (CMG: *n* = 28), a happy music group (HMG: *n* = 30), and a sad music group (SMG: *n* = 27). Negative emotions were induced in all participants using a COVID-19 video, followed by the music intervention condition. Participants underwent the face–word Stroop tasks during which event-related potentials (ERPs) were recorded. The N2, N3, P3, and late positive component (LPC) were investigated. The results showed that calm music and happy music effectively regulate young adults’ induced negative emotions, while young adults experienced more negative emotions when listening to sad music; the negative mood states at the post-induction phase inhibited the reaction of conflict control in face–word Stroop tasks, which manifested as lower accuracy (ACC) and slower reaction times (RTs). ERP results showed negative mood states elicited greater N2, N3, and LPC amplitudes and smaller P3 amplitudes. Further studies are needed to develop intervention strategies to enhance emotion regulation related to COVID-19 for other groups.

## 1. Introduction

Emotion regulation plays an important role throughout life. Effective emotion regulation underpins the ability to maintain physical and mental health and sustain individual development [1]. Individuals experiencing sustained stress over long periods are susceptible to negative emotions; negative stress reactions seriously affect physical and mental health, ultimately leading to mental health problems [2,3,4]. Music is an important means of emotion regulation [5]. Music contributes to “cultivating emotions that are helpful—and managing emotions that are harmful” and, as such, “it is one of the central concerns of the field of emotion regulation” [6] (p. 1). Previous studies have shown that both mindfulness meditation and effective music listening have significant positive effects on emotion regulation [7,8,9,10].

According to stress and coping theory [11], based on their own cognitive evaluation, individuals can suffer a series of emotional, behavioral, physiological, and psychological stresses when encountering crises. Stress during a crisis can cause changes in the autonomic nervous and neuroendocrine systems, produce strong negative emotional experiences, and even stimulate defensive behavior to manage sudden threats, which has adaptive significance [12,13,14]. Since its discovery in December 2019 and the outbreak in Wuhan (Hubei Province, China) in February 2020 [3,4], the coronavirus disease 2019 (COVID-19) has been spreading globally [15]. COVID-19 has an important impact on the public’s mental health [4,16,17]. Previous studies have shown that people may experience mood disturbance and stress [2,3]; increased sensitivity to social risks [3,4]; and negative emotions, such as anxiety, depression, anger, helplessness, and panic, against the background of the major epidemic of COVID-19 [2,4,18] conducted a survey of stress perception and sleep quality among 1630 healthy individuals in 32 provinces and cities of China from February 18 to February 25, 2020. The results found that during the COVID-19 outbreak, more than one-third of the general population had poor sleep quality due to increased perceived stress. This finding demonstrated that the outbreak of COVID-19 has also resulted in adverse stress effects in healthy people, such as heightened anxiety and depression, due to their disrupted working lives [18,19].

Diamond (2003) pointed out that “the optimal developmental outcome with respect to emotion regulation is not affective homeostasis, but rather a dynamic flexibility in emotional experience” (p. 1). Music listening, as a dynamic process of emotional expression, plays an important role in the regulation of negative emotions [20,21]. However, several studies have shown that “music was not a magic pill that could immediately resolve a negative mood and nor was it always helpful” [22] (p. 9). Unhelpful listening habits are not conducive to regulating negative emotions; only the effective strategies of listening to music, such as self-chosen music [20], self-awareness, and conscious music listening choices [22], can help regulate daily emotions as well as induced negative emotions. Effective music listening can reduce negative emotional experiences, improve psychological health [10,21,23,24,25,26,27], and enhance cognitive functions [10,20,21].

Effective emotion regulation strategies are closely related to individual attention allocation [12,24,28]. Attention training is closely related to emotion regulation, and studies have shown that mindfulness-based attention training can effectively reduce negative emotions and enhance well-being [29,30,31,32,33]. Mindfulness meditation, as an effective strategy for emotion regulation, can effectively increase the level of attention and improve negative emotions [8,29,32,34]. Temporary mindfulness meditation training, as a relaxed state of enhanced self-regulation, effectively regulated individuals’ negative moods and non-judging of inner experience, which improved emotion processing [35,36,37,38]. Mindfulness-based music listening has been shown to regulate negative mood [10,39,40], decrease psychological stress and anxiety symptoms [40], and enhance body awareness [30] and attentional control [10].

Attention plays an important role in conflict control [41,42,43,44]. Conflict control, which reacts differently to conflicting information, is the ability of the brain to monitor conflicts in the process of information processing and measures the ability of inhibitory control at the cognitive level [45,46]. The Stroop task, which is generally used to detect the ability of cognitive conflict, is a classic experimental paradigm that investigates behavioral and neural correlates of emotion–cognition interaction and can reveal the mechanism of cognitive conflict [41,44,47,48]. By using stimuli with different emotional valences, the Stroop task can effectively explore conflict detection and resolution [44]. The affective Stroop task “is an indicator of cognitive control and enables the quantification of interference in relation to variations in cognitive load” [44](p. 1). The task accuracy (ACC) and reaction times (RTs) in congruent or incongruent Stroop trials during the Stroop task performance reflect cognitive control mechanisms [41,44,49]. Relying on the technical means of event-related potentials (ERPs), the Stroop task can also be also used to investigate the neural correlates of emotion–cognition interaction [41,42].

In conflict control, individual emotional states are closely related to attention bias. Hence, attention training strategies can induce attention shifts and regulate emotions. An increasing number of researchers are focusing more on the effects and the attention mechanism in the process of emotion regulation [5,14,22,30,43,50,51,52]. Some studies have shown that attention shifts can effectively reduce the level of brain activation related to emotions [8,41,44,48,53]. Neurophysiological evidence has shown that different attentional biases [54] affect the intensity and depth of individual emotional experiences [55] and determine the effect of emotional regulation [5,12,26].

Attention also plays an important role in the emotional experience induced by music [5,30,56]. The neurophysiological evidence regarding sad music shows differences in attentional bias-activated circuits and physiological responses in music-related brain regions, including the frontal, central, and parietal regions [23,57]. Research on the influence of personal emotional state on attention processing shows that the characteristics of individual attention processing are influenced not only by emotional stimuli [41], but also the individual’s emotional state [43,44]. Previous studies typically induced specific emotional states using various methods and technical means [20,43], followed by an examination of individuals’ attention characteristics when in different emotional states, in response to emotional stimuli [41,44,48].

Relying on the technical means of neuroscience, it is possible to discover the differences in the activation of the individual’s neurophysiological system during the process of emotional regulation and attention distribution and then to reveal the factors influencing this process [43,58,59]. As the immediate processing of emotion regulation is a dynamic process [48,60], ERPs have unique advantages in exploring the effects of individual brains on emotion regulation [52,61]. Previous ERP studies [50,62,63,64] used conflict control tasks to explore conflicting decision-making and found that N2, N3, and P3 were important electroencephalography (EEG) indicators related to attention core processing. In attentional control, N2 is related to attentional bias during the early processing of attention. It has a negative potential approximately 200–350 ms after the appearance of stimulus; it reflects cognitive control in the visual modality, involving frontocentral and parietal scalp distributions [65]. Most stimulus-driven affective ERP modulations can be generated automatically in conditions where the participant is viewing the images [66].

Another useful variable for studying emotional processing to visual stimuli is N3 [63,67]. N3 is a negative potential occurring approximately 250–350 ms after the appearance of stimulus [62]. Violations in semantic integration of conflict control are associated with N3 [62]; this reflects pre-semantic perceptual processes [68,69], as well as biasing perceptual processes toward contextual information [70,71]. According to object identification theories, greater N3 amplitudes indicate the initial categorization of object identification [62,63,64]. N3 effects reflect the processing of object identification and the categorization of all semantic matches and mismatches [63,69]. Previous studies reported that the P3 is conceptualized as a direct index of response inhibition and interference [43,72]. Related to higher-order cognitive resources [42,43,50], P3 is a positive potential occurring approximately 300–600 ms after stimulus appearance [73]. The smaller P3 amplitudes, as well as attenuated P3 amplitudes in the conditions, are related to the cognitive control of task-independent interference information and reflect increased cognitive control [43,72]. The results of previous studies [74,75,76] show that negative stimuli elicited larger P3 amplitudes than neutral stimuli but were incongruent with the evidence from covert emotional studies showing that negative stimuli elicited smaller P3 amplitudes than did neutral stimuli [72]. This result may be caused by the choice of experimental methods (covert vs. overt), participants’ negativity bias to emotional stimuli, and inhibitive process, which most likely contributes to the smaller P3 amplitudes under negative conditions [72].

Additionally, individuals with negative mood states elicited greater late positive component (LPC) amplitudes than those with neutral or positive mood states [43]. The LPC, as a positive potential occurring approximately 600–1000 ms after the stimuli appearance [42,58], is considered typical in EEG studies of emotion regulation [52]. The LPC and LPP (late positive potential) represent the same ERP component. Many studies have found that LPC amplitudes can reflect the arousal of emotional stimuli [77], the individual attention to emotional stimuli [78], and the regulation of emotional stimuli in the process of emotion regulation [52,53,79], which is reflected in the regulation effect of emotion regulation [52,80]. Increased LPC amplitudes reflect more attentional resources in the processing of stimuli involving motivation or emotion [58,81]. Moreover, LPC is related to the engagement of controlled cognitive resources that reflect higher-order cognitive processes, such as response decisional processing [82]. The LPC amplitude evoked by negative pictures was larger than that evoked by positive and neutral pictures, suggesting that the negativity bias also occurred in the later evaluation stage of emotion processing [74].

Although previous studies [2,4,16,19] have shown that healthy individuals are prone to stress responses and induced negative emotions, the impact of negative emotions induced by COVID-19 on the neural markers of cognition is unclear. In addition, to date, researchers have not examined whether ERPs associated with different mood states (e.g., baseline, post-induction, and post-intervention) differ among subgroups. Given that the different temporal stages of attention processing are affected by negative mood states [43,74], it is plausible that underlying neurophysiological differences are also present between these subgroups. To evaluate this premise, different mood state effects of the baseline, post-induction, and post-intervention phases on conflict control and ERPs were assessed among a calm music group (CMG), a happy music group (HMG), and a sad music group (SMG). A revised face–word Stroop research paradigm tapping face-specific conflict control [41,44] was used in concert with negative mood induction (i.e., sadness) via exposure to mood states induced by video clips related to COVID-19. As previous ERP studies of conflict control and attentional distribution emphasized N2, N3, P3, and LPC components, we also adopted these ERPs as the focus of group and mood state comparisons [41,43,44,67]. Based on the assumption that coping with negative mood states can lead to conflict in completing Stroop tasks, we hypothesized that negative mood states would elicit greater N2, N3, and LPC amplitudes and smaller P3 amplitudes than neutral or positive mood states at the baseline and post-intervention phases, especially among the CMG and HMG, compared to the SMG. The aim of the current study was to determine the behavioral and neural correlates of listening to music based on mindfulness meditation to regulate negative emotions related to COVID-19. The revised face–word Stroop task was used to explore the attentional distribution and cognitive conflict in different mood states. The within-subject (baseline, post-induction, post-intervention) and between-subjects (CMG, HMG, and SMG) differences in ERPs were investigated to illustrate neural mechanisms underlying attentional distribution and the cognitive conflict of negative emotion related to COVID-19. Based on previous studies, we hypothesized the following.
First, the post-induction phase will exhibit lower ACC and slower RTs than the baseline and post-intervention phases in within-subjects conflict control performance on the face–word Stroop task.Second, in the early processing of cognitive conflict, attentional bias under induced negative mood states will allocate more cognitive resources to attentional responses of face-related stimuli. This will be reflected in greater N2 amplitudes at post-induction, and for the SMG in the post-intervention phase, compared to the baseline phase, and the HMG in the post-intervention phase.Third, the assumption that coping with negative mood states can lead to conflict in the face–word Stroop tasks will be reflected in greater N3 and smaller P3 amplitudes at post-induction, and post-intervention for the SMG, compared to the baseline, and HMG at the post-intervention phase.Finally, participants who completed the face–word Stroop task will be able to conduct higher-order cognitive processes in negative mood states. This will be reflected in enhanced LPC amplitudes at post-induction and for the SMG in the post-intervention phase, compared to the baseline and the HMG post-intervention.

## 2. Methods

### 2.1. Participants

The healthy participants were recruited through campus advertisements (N = 85, 68.24% females, M = 20.69, SD = 1.13) and were required to abstain from taking substances or medications that could potentially influence their concentration. Additionally, they were required to disclose any history of major psychological disorders. All participants reported being right-handed and having normal hearing and speech and normal or corrected-to-normal vision. Before starting the experiment, all participants read the instructions and asked questions about the experiment before giving written consent to participate. This study was approved by the Southwest University Ethics Committee (IRB No. H19072).

### 2.2. Stimuli

#### 2.2.1. Material Evaluation

In the current study, participants rated the video’s emotional content using a 100 mm visual analog scale (VAS) (not at all sad–very sad; not at all tense–very tense) at the end of the video. The emotional content of musical stimuli was rated by participants using 100 mm VAS (calm music: not at all calm–very calm; happy music: not at all happy–very happy; sad music: not at all sad–very sad) at the end of the musical listening [83].

#### 2.2.2. Video of Experimental Simulation

The video of experimental simulation used to induce participants’ negative emotions in this study is derived from a real event related to COVID-19 from China in 2020. The stimulus material was made with the specialized video editing software “iJianJi,” and the duration of the video is 4 min. In the current study, the emotional content of the video stimuli was negative emotion (sad: M = 71.03, SD = 19.43; tense: M = 69.12, SD = 22.56). The video in this study is a combination of different video clips related to COVID-19 outbreak in China from February to May, 2020, these video clips are selected from the website of Baidu Haokan APP (https://haokan.baidu.com/, accessed on 10 September 2020).

#### 2.2.3. Mindfulness Meditation Audio

The Chinese version of the mindfulness meditation script used in this study was derived from the English version of a mindfulness script [84]. The translation of the text was proofread and revised by two graduate students majoring in English. The audio was recorded by professionals who had been trained in meditation and yoga for 10 years in a soundproof room using the specialized recording software Xunjie audio recorder. The duration of the audio recording was 10 min, and it was recorded in MP3 format [85].

#### 2.2.4. Musical Stimuli

The stimuli set consisted of three Chinese classical folk instrumental music works, which were taken from the commercially available “Kugou” music software (Version 9.1.32MAC, Guangzhou Kugou Computer Technology Co., Ltd. (Guangzhou, China), www.kugou.com, accessed on 11 September 2019), which is a professional online music player application. These high-quality music works included three emotion levels, namely calm, happy, and sad, and the duration of each piece of music is approximately 3 min and 20 s. The emotional valence of these music materials was assessed by 50 musicians using a 9-factorial GEMS model [86,87]. In the current study, calm, happy, and sad music had Cronbach’s alpha values of 0.83, 0.86, and 0.88, respectively. The participants reported that all of the musical stimuli were unfamiliar. In the current study, the emotional contents of musical stimuli were calm music (M = 75.03, SD = 20.58), happy music (M = 80.33, SD = 22.38), and sad music (M = 71.61, SD = 19.52).

### 2.3. Self–Reported Measures

#### 2.3.1. The Positive and Negative Affect Schedule (PANAS)

The Positive and Negative Affect Schedule (PANAS) [88] is a 20-item questionnaire that assesses participants’ current mood state in terms of negative and positive affect. Participants rated the extent to which each of the 20 adjectives described their current feeling on a 5-point scale ranging from 1 (*very slightly or not at all*) to 5 (*extremely*). Scores for this scale were summed separately for the positive and negative affect. As originally reported, Cronbach’s alphas of positive affect (PA) ranged from 0.86 to 0.9, and those of negative affect (NA) ranged from 0.84 to 0.87 [88]. In the current study, PA had a Cronbach’s alpha of 0.81 and NA had a Cronbach’s alpha of 0.80. The PANAS was used to assess participants’ mood state in three stages: baseline, post-induction of negative emotions induced by the video, and post-intervention of mindfulness-based music listening.

#### 2.3.2. The Toronto Mindfulness Scale (TMS)

The Toronto Mindfulness Scale (TMS) is a promising measure of the mindfulness state with good psychometric properties and is predictive of treatment outcomes [89]. The Chinese version of the TMS, which was revised by Chung and Zhang [90], has 13 items [89] and uses a 5-point scale ranging from 0 (*not at all*) to 4 (*very much*). TMS is a widely used instrument of state mindfulness, and the higher the scores, the higher the mindfulness state [49,89]. As originally reported, the TMS includes two factors: curiosity and decentering. The Cronbach’s alphas of curiosity ranged from 0.62 to 0.82, while those of decentering ranged from 0.56 to 0.78. In the current study, TMS had a Cronbach’s alpha of 0.81 and was used to measure participants’ mindfulness meditation state before and after post-intervention of mindfulness-based music listening.

### 2.4. The Face–Word Stroop Task

Using the revised face–word Stroop task [91,92,93], this study investigated the effects of cognitive control and attentional distribution in different mood states. Polarized emotional valence (positive-negative) of the stimuli [67] and picture stimuli in congruent and incongruent contexts [42,64,69] were considered to explain the emotional meaning in the processing of visual stimuli. Therefore, the materials of the face–word Stroop task comprised 20 happy pictures of different adult faces (10 female and 10 male adult pictures), 20 sad pictures of different adult faces (10 female and 10 male adult pictures) selected from the Chinese Affective Picture System [94], and emotional words (“高兴”: happy, and “悲伤”: sad). Our revised face–word Stroop task included 80 trials and four conditions of stimulus: sad congruent, sad incongruent, happy congruent, and happy incongruent. The combinations of the four conditions in the face–word Stroop task were as follows: sad congruent (20 trials; the combination of a sad face and the word “悲伤”), sad incongruent (20 trials; the combination of a sad face and the word “高兴”), happy congruent (20 trials; the combination of a happy face and the word “高兴”), and happy incongruent (20 trials; the combination of a happy face and the word “悲伤”).

In the face–word Stroop tasks, a fixation appeared for 500 ms, and a stimulus was then presented on the monitor until the participant responded. If the participant did not respond, stimuli would automatically disappear after 1000 ms, followed by an inter-stimulus interval of 500 ms (Figure 1). The stimuli were presented randomly and repeated twice. Participants were asked to press the button “1” for congruency and to press the button “2” for incongruency as well as to press a button as soon as possible during the Stroop task. The task order was counterbalanced across participants.

The Stroop task consisted of one practice block of 80 trials followed by 320 trials presented in a completely random order. Each picture was identical in size (300 × 260 pixels), resolution (96 dots per inch), brightness, and background. Stimuli were presented on a 19-inch Lenovo computer monitor, with the center of the screen set at eye level. Participants were instructed to remain as still as possible and minimize their eye blinking to reduce experimental artifacts in the EEG data collection.

### 2.5. Procedure

Participants were informed that this study was about attention to the face–word Stroop task and that they would perform tasks on a computer. The experiment was divided into three sections: baseline, post-induction, and post-intervention Stroop tasks. In the baseline phase, after providing written consent, participants rated their baseline mood state level using the PANAS, reported their state of mindfulness meditation using the TMS, and completed the Stroop task. In the post-induction phase, a negative mood state was induced in all participants using a video related to COVID-19, after which they completed the Stroop task and then rated their mood state using the PANAS. In the post-intervention phase, participants performed the Stroop task after mindfulness-based music listening, aimed at regulating their negative emotions using Chinese instrumental music with different valences. Finally, participants rated their post-intervention mood state level using the PANAS and reported their state of mindfulness meditation using the TMS. The Stroop tasks were presented in a counterbalanced order among the participants. The EEG data were recorded throughout the entire experiment.

Based on differences in emotional valence, 90 participants were randomly divided into three groups (CMG, HMG, and SMG), with 30 participants per group. In the post-intervention phase, the participants received the intervention of the entire music listening process based on mindfulness meditation. The duration of the full experiment was approximately 50 min.

### 2.6. Behavioral Analyses

Repeated-measures analysis of variance (ANOVA) was conducted to identify between-group differences in age, PANAS score, and TMS score. Repeated-measures ANOVAs (3 (groups: CMG, HMG, and SMG] × 3 (measure: baseline, post-induction, post-intervention) × 2 (emotional valence: happy face, sad face) × 2 (condition: congruent, incongruent)) were conducted for the ACC and RTs of emotional faces in the face–word Stroop task, with group as a between-subjects factor and measure, emotional valence, and condition as within-subject factors. The analyses were conducted using SPSS 22.0. The *p*-values were adjusted for sphericity using the Greenhouse–Geisser method. Post-hoc *t*-tests were performed using Bonferroni adjustments for multiple comparisons.

### 2.7. EEG Recording and Analyses

Brain electrical activity was recorded from 32 scalp sites using tin electrodes mounted in an elastic cap (Neuroscan, Charlotte, NC, United States), with the reference electrodes placed on REF (fronto-central aspect) and a ground electrode on the medial frontal aspect (GRD). The vertical electrooculogram (IO) was recorded with an electrode placed infraorbitally near the left eye. All inter-electrode impedance was maintained below 5 kΩ. Data processing was performed with MATLAB R2014a using the EEGLAB toolbox 14.1.1b.

Individual and grand ERP averages were created for emotional face stimuli, and the resulting grand averages were based on the correct trials. We first downsampled the data from 1000 to 256 Hz and performed high-pass filtering at 0.1 Hz and low-pass filtering at 45 Hz. We selected the left and right mastoids as the reference sites. Data were epoched from 200 ms prior to stimulus onset to 1000 ms after presentation and were baseline-corrected to the pre-stimulus interval. Trials with electroculogram (EOG) artifacts (ocular movements and eye blinks), artifacts because of amplifier clipping, bursts of electromyographic activity, or peak-to-peak deflections exceeding ±80 μV were excluded from averaging before independent component analysis (ICA). The components, including EOG artifacts and head movement, were removed from the results of the ICA results after visual inspection. Based on previous studies [46,95,96,97], the topographical distribution of the grand-averaged ERP activities, the ERP components, and their time epochs were as follows: N2 (200–260 ms), N3 (270–340 ms), P3 (340–450 ms), and LPC (500–996 ms).

The following regions of interest (ROIs) (Figure 2) were selected [42,46]: frontal (F3, Fz, F4), frontal–central (FC3, FCz, FC4), central (C3, Cz, C4), central–parietal (CP3, CPz, CP4), and parietal (P3, Pz, P4). For the face–word Stroop task, repeated-measures ANCOVA (3 (Group: CMG, HMG, SMG) × 3 (measure: baseline, post-induction, post-intervention) × 2 (emotional valence: happy face, sad face) × 2 (condition: congruence, incongruence) × 5 (ROIs: frontal; frontal-central; central; central-parietal; parietal)) were conducted on the amplitudes of N2, N3, P3, and LPC, with group as a between-subjects factor and measure, emotion valence, condition, and ROIs as within-subjects factors. All analyses were conducted using SPSS 22.0. The *p*-values were adjusted for sphericity using the Greenhouse–Geisser method. Post-hoc *t*-tests with Bonferroni adjustments were used for multiple comparisons. We conducted outlier analyses on EEG data using ± 3 SDs, and five participants (three participants in CMG and two participants in SMG) were excluded from the study.

## 3. Results

### 3.1. Self-Reported Results

The participants’ demographic information and self-reported results are shown in Table 1. There were no significant between-group differences for age or sex (all *ps* > 0.05).

### 3.2. Questionnaire Results

Repeated-measures ANOVA on PANAS scores (Figure 3) showed a main effect of PA (*F* (2, 82) = 26.10, *p* < 0.001, $\eta_{p}^{2}$ = 0.24) and NA (*F* (2, 82) = 54.21, *p* < 0.001, $\eta_{p}^{2}$ = 0.4). The post-hoc *t*-test showed that post-induction scores were lower for PA and higher for NA than at the baseline and post-intervention (*p* < 0.05). Moreover, there were no significant differences between the baseline and post-intervention (*p* > 0.05). There was an interaction between NA and the group (*F* (4, 164) = 3.33, *p* = 0.02, $\eta_{p}^{2}$ = 0.08). A simple effect analysis showed that the score of the CMG was higher than that of the HMG and the SMG with no significant difference between the HMG and the SMG at the baseline and post-induction (*p* < 0.05). The scores of the SMG were higher than those of the CMG and the HMG (*p* < 0.05) with no significant difference between the CMG and the HMG in the post-intervention (*p* > 0.05). Additionally, there was no interaction of PA and group (all *ps* > 0.05).

Repeated-measures ANOVA showed that the TMS (Figure 4) had a main effect on mindfulness state: *F* (1, 82) = 69.95, *p* < 0.001, and $\eta_{p}^{2}$ = 0.46. The post-hoc *t*-test showed that post-intervention score was higher than the baseline score (*p* < 0.05). There were no main effects of the group and no interaction between mindfulness state and the group (all *p*s > 0.05).

### 3.3. Behavioral Results

Repeated-measures ANOVA on the ACC of emotional faces showed a main effect of measure (Table 2 and Figure 5), with *F* (2, 82) = 8.79, *p* < 0.001, and $\eta_{p}^{2}$ = 0.10, and the post-hoc *t*-test found that the ACC of post-induction was lower than that of the baseline and post-intervention (*p* = 0.02), the ACC of post-intervention was higher than that of the baseline (*p* = 0.05). We found no main effect of emotional valence and no interaction between measure and emotion valence on the ACC of emotional faces (*p* > 0.05). There was a main effect of condition, with *F* (1, 82) =85.13, *p* < 0.001, and $\eta_{p}^{2}$ = 0.51, and the post-hoc *t*-test found that the ACC of incongruence was lower than that of congruence (*p* < 0.001). There was an interaction between measure and condition (*F* (2, 82) = 5.00, *p* < 0.01, $\eta_{p}^{2}$ = 0.06) and simple effect analysis found that the ACC of incongruence was lower than that of congruence in all measures (*p* = 0.02). There was no main effect of the group and no interaction between measure and group (all *ps* > 0.05).

Moreover, repeated-measures ANOVA on RTs (Figure 6) for emotional faces showed a main effect of measure, (*F* (2, 81) = 15.62, *p* < 0.001, $\eta_{p}^{2}$ = 0.16) and the post-hoc *t*-test found that RTs at post-intervention were faster than those at the baseline and post-induction (*p* < 0.001). There was no significant difference between the baseline and post-induction (*p* > 0.05). There was a main effect of emotional valence (*F* (1, 82) = 104.36, *p* < 0.001, $\eta_{p}^{2}$ = 0.56), and simple effect analysis showed that RTs of happy faces were slower than those of sad faces (*p* < 0.001) with no interaction between measure and emotional valence observed (all *ps* > 0.05). There was a main effect of the condition (*F* (1, 82) = 269.29, *p* < 0.001, $\eta_{p}^{2}$ = 0.77), and simple effect analysis showed that RTs of incongruence were slower than those of congruence (*p* < 0.001). No interaction between measure and condition was found (all *ps* > 0.05). No main effect of the group and no interaction between measure and group were recorded (all *ps* > 0.05).

### 3.4. The ERPs Results

Grand average ERPs for the Stroop task of N2, N3, P3, and LPC at FCz are shown in Figure 7.

#### 3.4.1. N2

In the case of N2, an interaction effect between ROIs and measure, *F* (2, 81) = 7.28, *p* = 0.001, $\eta_{p}^{2}$ = 0.08, and simple effect analysis showed that the baseline and post-induction at Pz were greater than post-intervention (*p* < 0.05), with no significant difference between the baseline and post-induction (*p* > 0.05). There was an interaction between measure and group, *F* (2, 82) = 3.53, *p* = 0.03, $\eta_{p}^{2}$ = 0.08; simple effect analysis showed that N2 mean amplitudes of CMG were greater than that of HMG in post-intervention (*p* = 0.01), whereas no such an effect was found between CMG and SMG or between HMG and SMG (all *ps* > 0.05). No significant between-group differences were found in the baseline and post-induction phases (all *ps* > 0.05).

There was an interaction effect of emotional valence and measure, *F* (1, 82) = 71.77, *p* < 0.001, $\eta_{p}^{2}$ = 0.47. Simple effect analysis showed that the N2 mean amplitudes of happy faces were greater than sad faces at three Stroop task measures (*p* < 0.001). There was an interaction effect between condition and measure, *F* (1, 82) = 8.91, *p* < 0.004, $\eta_{p}^{2}$ = 0.10, and simple effect analysis showed that the N2 mean amplitudes for incongruence were greater than for congruence at post-induction and post-intervention (all *ps* < 0.01), whereas no such effect was found at the baseline (*ps* > 0.05). There was a main effect of ROIs, *F* (4, 79) = 11.85, *p* < 0.001, $\eta_{p}^{2}$ = 0.38; a post-hoc test showed that N2 mean amplitudes were greatest in Fz, and the magnitude order of N2 mean amplitudes was Fz > FCz > Cz > CPz > Pz.

#### 3.4.2. N3

There was an interaction effect for ROIs and measure, *F* (2, 81) = 5.97, *p* = 0.001, $\eta_{p}^{2}$ = 0.07, and simple effect analysis on the N3 mean amplitudes of Pz showed that post-induction was greater than post-intervention (*p* < 0.05), with no significant difference between the baseline and post-induction recorded (*p* > 0.05). There was an interaction between measure and group, *F* (2, 82) = 3.53, *p* = 0.03, $\eta_{p}^{2}$ = 0.08, and simple analysis showed that CMG and SMG were greater than HMG at the baseline (*p* = 0.012); no such an effect was found between CMG and SMG (*p* > 0.05). There were no significant between-group differences at post-induction (all *ps* > 0.05); N3 mean amplitudes of CMG were greater than that of HMG in intervention (*p* = 0.012), no such effect was found between CMG and SMG or between HMG and SMG (*p* > 0.05).

There was an interaction effect for emotional valence and measure, *F* (1, 82) = 5.69, *p* < 0.02, $\eta_{p}^{2}$ = 0.07, and simple effect analysis showed that the N3 mean amplitudes for the happy face was greater than for the sad face at post-induction (*p* < 0.001), while no such effect was found at the baseline or post-intervention (all *ps* > 0.05). The interaction effect between condition and measure was marginally significant in post-induction, *F* (1, 82) = 3.44, *p* < 0.067, $\eta_{p}^{2}$ = 0.04, and simple effect analysis showed that the N3 mean amplitudes of incongruence were greater than that of congruence at post-induction (all *ps* < 0.01), whereas such an effect was not found at the baseline or post-intervention (all *ps* > 0.05). Additionally, there was a main effect of ROIs, *F* (4, 79) = 28.73, *p* < 0.001, $\eta_{p}^{2}$ = 0.59, for which a post-hoc test showed that N3 mean amplitudes were greatest in Fz. The magnitude order of N3 mean amplitudes was Fz > FCz > Cz > CPz > Pz.

#### 3.4.3. P3

Regarding P3, a main effect of the measure was recorded (*F* (2, 82) = 4.38, *p* = 0.01, $\eta_{p}^{2}$ = 0.05), and the post-hoc *t*-test on measure showed that P3 mean amplitudes at the baseline were greater than those at post-induction and post-intervention (*p* = 0.04). An interaction between ROIs and measure was observed (*F* (2, 81) = 3.56, *p* = 0.012, $\eta_{p}^{2}$ = 0.04), and simple effect analysis showed that P3 mean amplitudes at post-induction and post-intervention at Fz, FCz, Cz, and Pz were smaller than that at the baseline (*p* < 0.05); no such effect was found at CPz (*p* > 0.05).

The interaction between measure and group at post-intervention was marginally significant (*F* (2, 81) = 2.65, *p* = 0.07, $\eta_{p}^{2}$ = 0.06), and simple effect analysis showed that P3 mean amplitudes of the CMG were greater than those of the HMG and SMG (*p* < 0.05), with no significant difference between the HMG and SMG. There were no significant between-group differences at the baseline or post-induction (all *p*s > 0.05). Additionally, a main effect of ROIs was observed, *F* (4, 79) = 96.92, *p* < 0.001, $\eta_{p}^{2}$ = 0.54; a post-hoc test showed that P3 mean amplitudes were greatest in Pz, and the magnitude order of P3 mean amplitudes was Fz < FCz < Cz < CPz < Pz.

#### 3.4.4. LPC

The main effect of the measure was observed (*F* (2, 81) = 4.67, *p* = 0.01, $\eta_{p}^{2}$ = 0.05), and the post-hoc *t*-test performed on measure showed that LPC amplitudes at post-induction were greater than those at the baseline and post-intervention (*p* = 0.23), with no significant difference between post-induction and post-intervention. We found an interaction between ROIs and measure (*F* (2, 81) = 4.67, *p* = 0.001, $\eta_{p}^{2}$ = 0.05). Simple effect analysis of Cz, CPz, and Pz showed that LPC mean amplitudes at the post-induction were greater than at the baseline and post-intervention, with no significant difference between the baseline and post-intervention (*p* < 0.05).

There was an interaction between measure and group at the baseline, *F* (2, 82) = 5.33, *p* = 0.007, $\eta_{p}^{2}$ = 0.12, and simple effect analysis showed that LPC mean amplitudes for the HMG and SMG were greater than for the CMG, with no significant difference between the HMG and SMG (*p* < 0.05). No interaction was observed at post-induction (all *ps* > 0.05); an interaction was found at post-intervention, *F* (2, 82) = 5.03, *p* = 0.009, $\eta_{p}^{2}$ = 0.11. Simple effect analysis showed that LPC mean amplitudes for the HMG and SMG were greater than those for the CMG, with no significant difference between the HMG and SMG (*p* < 0.05).

A main effect of emotional valence was recorded, *F* (3, 80) = 6.45, *p* = 0.013, $\eta_{p}^{2}$ = 0.07, and the post-hoc *t*-test on stimulus showed that LPC mean amplitudes for sad faces were greater than for happy faces (*p* = 0.01). A main effect of condition was recorded, *F* (3, 80) = 15.71, *p* < 0.001, $\eta_{p}^{2}$ = 0.16, and the post-hoc *t*-test on stimulus showed that LPC mean amplitudes of incongruence were greater than of congruence (*p* = 0.001). There was a main effect of ROIs, *F* (4, 79) = 26.96, *p* < 0.001, $\eta_{p}^{2}$ = 0.58, and the post-hoc *t*-test showed that LPC mean amplitudes were greatest in Cz and smallest in Pz.

## 4. Discussion

In our novel examination of the behavioral and ERP correlates of mindfulness-based music listening regulating negative emotions related to COVID-19, the hypotheses were partially confirmed. The findings indicate that (1) calm music and happy music effectively regulated young adults’ negative emotions induced by the video related to COVID-19, (2) young adults experienced more negative and less positive emotions while listening to sad music, (3) mindfulness meditation effectively promoted the physical and mental relaxation of young adults, and (4) the post-induction phase exhibited lower ACC and slower RTs than the baseline and post-intervention phases in within-subjects conflict control performance in the face–word Stroop task, and ERP results showed that conflict control towards incongruent emotional face stimuli occupied more higher-order cognitive resources in negative mood states than at the baseline and post-intervention. The Stroop task’s performance (lower accuracy and longer reaction time) was influenced by the short 8-min video related to COVID-19 in the healthy young adults, which may suggest that long-term exposure to COVID-19 may not only induce negative emotions [18], but also have a negative effect on cognition.

Our results for behavioral tasks, consistent with those of previous studies [20,22,98], indicate that mindfulness-based listening to music with positive valence effectively regulated induced negative emotions related to COVID-19. TMS results showed that mindfulness meditation training effectively improved the maintenance of a relaxed state during music listening. Consistent with previous studies [41,43,99], our behavioral results showed that compared to the baseline and post-intervention, induced negative mood states at the post-induction phase exhibited lower ACC and slower RTs in the face–word Stroop task. Compared to the post-induction phase, no significant between-group differences in the ACC and RTs of the baseline and intervention phases were found, indicating that mindfulness-based listening to music with positive valence regulated induced negative emotions [20], and the processing of attention attentional distribution in conflict tasks was affected by individual emotional states.

In the performance of the Stroop task, compared to the baseline and post-intervention, the negative mood states of the post-induction phase inhibited the reaction of conflict control, as evidenced through lower ACC and slower RTs. Moreover, the ERP evidence also supported the behavioral results. First, ERP results showed that young adults with negative mood states elicited greater N2 and N3 amplitudes at post-induction and post-intervention for the CMG compared to the HMG in the early processing of conflict control. Second, the P3 of amplitudes at post-induction and post-intervention phases were smaller than at the baseline. In the post-intervention phase, P3 amplitudes of CMG were greater than those of the HMG and SMG. Finally, compared to the baseline and post-intervention, significantly greater LPC was elicited at post-induction in all sub-groups. LPC amplitudes for the HMG and SMG were greater than those of the CMG at the baseline and post-intervention in the face–word Stroop task. Notably, although no significant between-group differences were found at the post-intervention phase in the ACC and RTs of the Stroop tasks, the SMG had higher NA and lower PA scores on the PANAS than the CMG and HMG. This finding indicates that more cognitive resources may be occupied in the performance of conflict tasks under negative and positive emotional states, induced by listening to sad and happy music, respectively.

Individual emotional states are closely related to attentional distribution in emotional regulation [7,48,74]. In the emotion–cognition interaction of conflict control, our ERP results supported the important role of attention distribution in the process of emotion regulation [43,46,50]. Consistent with previous studies [43,48,54,58,63,67], the induced negative mood states in the post-induction phase elicited greater N2 and N3 amplitudes in the early processing of conflict control. This shows that, compared with the baseline phase, induced negative mood states in post-induction occupied more cognitive resources to complete the face–word Stroop task. In our study, N2 and N3 amplitudes were significantly greater at post-induction than at post-intervention in the face–word Stroop tasks. N2 and N3 effects suggested that young adults with induced negative mood at the post-induction experienced more cognitive conflict than young adults with positive mood states at post-intervention when completing the face–word Stroop task. The results show N2 to be an ERP component related to attentional bias [65]; N3 was a variable for studying emotional processing of visual stimuli under different emotional states during the early processing of attention [63]. N2 and N3 both reflected the neural activation modality of the parietal region in emotion regulation [65].

Although the P3 or the LPC signals the cognitive evaluation of the stimuli’s meaning [72,74,75], their role in emotion regulation is different. In our study, P3 amplitudes in the parietal region were smaller at post-induction and post-intervention than at the baseline phase in the face–word Stroop task, indicating that P3 amplitudes under mood states induced by the video and mindfulness-based music listening decreased significantly compared to neutral mood state at the baseline. Our P3 results are consistent with previous findings [43,72,100]; the smaller P3 amplitudes are related to conflict control of task-independent interference information [43,100]. This may be caused by expression suppression, which elicits a smaller P3 amplitude in the parietal region under induced mood states [43,72]. Our results indicate that P3 is an effective ERP component for suppressing emotional expression behavior [43,72].

Increased LPC magnitudes in the central, central–parietal, and parietal regions reflect more attentional resources put toward stimuli processing [72,74,75], which may involve motivation or emotion [81]. This means that LPC amplitudes depend on the distribution of cognitive resources; in other words, the more cognitive resources the task needs, the greater the LPC amplitudes [66]. The ERP results suggest that significantly greater LPC amplitudes were elicited at post-induction compared with the baseline and post-intervention. Additionally, the LPC amplitudes of the HMG and SMG were greater than those of the CMG at post-intervention. Our findings illustrate that higher LPC amplitudes in the central, central–parietal, and parietal regions reflect the increased occupation of higher-order cognitive resources in negative mood states while completing Stroop tasks. These results dovetail with existing evidence [42,82] that LPC is an effective ERP index for the detection of emotional arousal in emotion regulation.

In summary, by taking all features of ERP components (N2, N3, P3, and LPC) into consideration in conflict control, we can comprehensively understand why mindfulness-based listening to music with positive emotional valences can effectively regulate young adults’ induced negative emotions related to COVID-19. Our results showed that the specific processing of emotional faces in conflict control is tightly related to the following ERP components: greater N2 is closely related to attentional bias in cognitive conflict [65], and N3 is closely related to violations in semantic integration of conflict control [62], which reflects pre-semantic perceptual processes to emotional visual stimuli during the early processing of attention [67,68,69]; P3 is related to congruent cognitive evaluation of the meaning of stimuli, and expression suppression in conflict control elicited a smaller P3 amplitude under a negative condition [43,72]; and LPC is an effective ERPs index for detection of emotional arousal in emotion regulation [58,82]. Additionally, we found that N2 and N3 amplitudes were greatest in the frontal region, compared to other regions; P3 amplitudes in the frontal, frontal–central, central, and parietal regions were decreased at post-induction and post-intervention, while LPC amplitudes in the central, central–parietal, and parietal regions were greater at post-induction than at the baseline and post-intervention. Consistent with previous studies [43,58,72], this result indicates apparent N2, N3, P3, and LPC components in the attentional processing of conflict control, and may provide a new direction for exploring attention training and emotion regulation strategies.

The main limitations of this study should be noted. First, although induced negative mood concerning COVID-19 induced by mindfulness-based music listening regulation is relevant and common among college-age adults, the findings of this study may not be generalizable to younger or older age groups, or to non-Chinese participants. These groups should be the focus of future studies. Second, although our results reflect possible neural responses in response to music listening with distinct emotional valences based on mindfulness meditation regulating induced negative mood, it is not clear whether the pattern of effects would extend to self-chosen sad music and vocal music works. Additionally, stress provoked by COVID-19 may bring about a variety of acute psychological or physical disorders, such as acute stress disorder and post-traumatic stress disorder and other serious stress reactions. This remains an avenue for future research and an important topic to explore how to regulate and alleviate these stress disorders through effective music listening based on mindfulness meditation. Finally, as noted above, specific music listening strategies are based on the effects of music listening and emotion regulation, which could effectively alleviate negative emotions and improve physical and mental health and well-being in the future.

## 5. Conclusions

The current study illustrates the effects of mindfulness-based music listening regulation on induced negative emotions in the course of conflict control tasks. The results showed that mindfulness-based listening to calm and happy music effectively regulated young adults’ negative emotions induced by a COVID-19 video. Young adults with negative mood states experienced more negative and less positive emotions, and their response conflict occupied more higher-order cognitive resources. Attention distribution toward incongruent emotional stimuli induced greater N2 and N3 amplitudes and smaller P3 amplitudes. Additionally, LPC amplitudes were significantly induced in negative mood states. Our study enriches the theoretical models of emotion regulation by providing neural markers for future studies. Hence, an important direction for future research is to develop intervention strategies that decrease negative emotions related to COVID-19 for other age groups.

## Figures and Tables

**Figure 1 ijerph-18-07063-f001:**
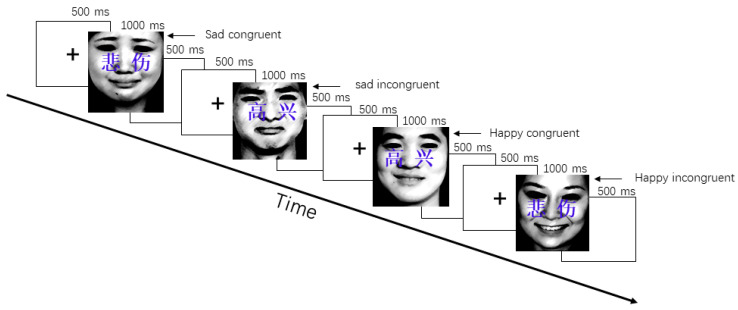
Example of the conflict paradigm. The flow-chart shows the time order of trials, which were completely randomized. The characters on the faces read sad (悲伤) and happy (高兴).

**Figure 2 ijerph-18-07063-f002:**
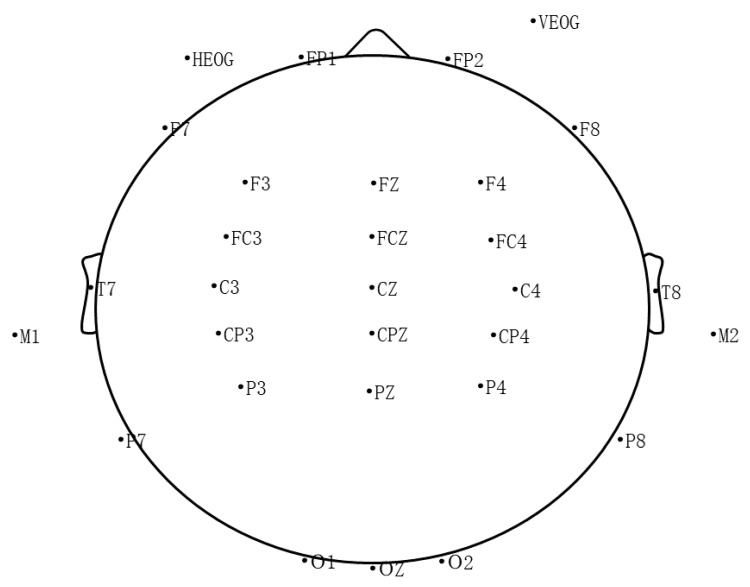
Plot of electrode sites (32 electrodes). The frontal (F3, Fz, F4), frontal–central (FC3, FCz, FC4), central (C3, Cz, C4), central–parietal (CP3, CPz, CP4), and parietal (P3, Pz, P4) were chose for EEG analysis.

**Figure 3 ijerph-18-07063-f003:**
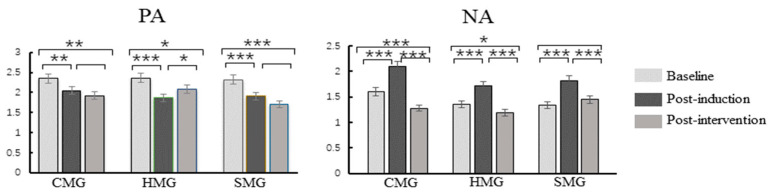
Positive and Negative Affect Schedule (PANAS) difference within-group before and after measure of the face–word Stroop task. PA: positive affect, NA: negative affect; CMG: calm music group, HMG: happy music group, SMG: sad music group; * *p* < 0.05, ** *p* < 0.01, *** *p* < 0.001.

**Figure 4 ijerph-18-07063-f004:**
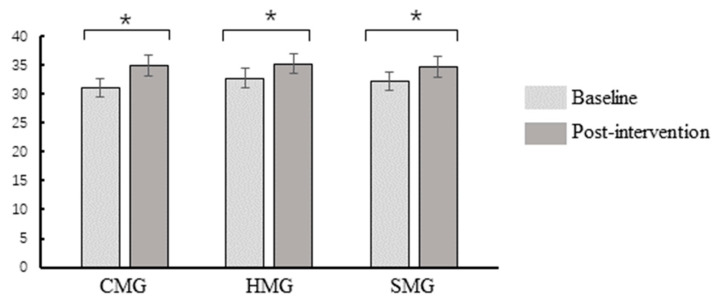
Toronto Mindfulness Scale (TMS) difference within-group before and after post-intervention. CMG: calm music group, HMG: happy music group, SMG: sad music group; * *p* < 0.05.

**Figure 5 ijerph-18-07063-f005:**
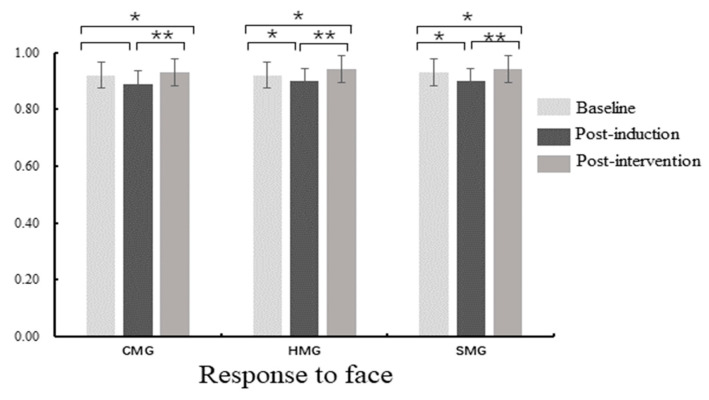
Accuracy (ACC) difference within-group in the face–word Stroop task; CMG: calm music group, HMG: happy music group, SMG: sad music group; * *p* < 0.05, ** *p* < 0.01.

**Figure 6 ijerph-18-07063-f006:**
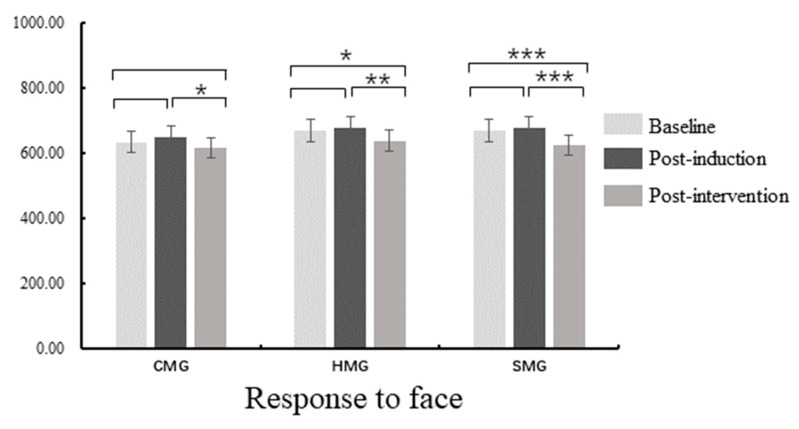
Reaction Times (RTs) difference within-group in the face–word Stroop task; CMG: calm music group, HMG: happy music group, SMG: sad music group; * *p* < 0.05, ** *p* < 0.01, *** *p* < 0.001.

**Figure 7 ijerph-18-07063-f007:**
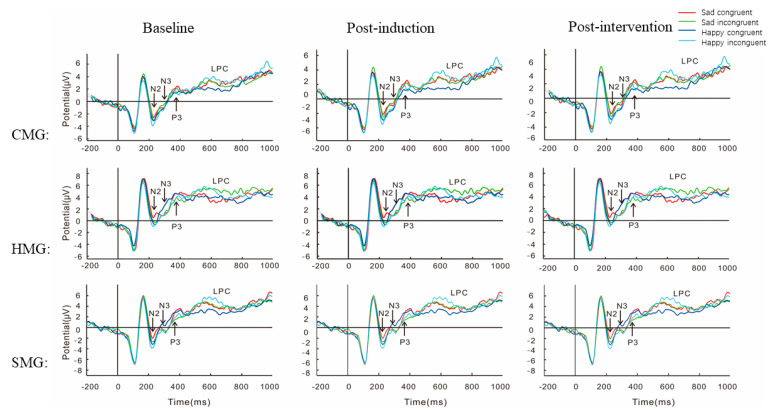
Grand average waveforms of N2, N3, P3, and late positive component (LPC) at site Fz in the face–word Stroop task. CMG: calm music group, HMG: happy music group, SMG: sad music group.

**Table 1 ijerph-18-07063-t001:** Participants’ demographic information and self-report results.

Variable		CMG (M ± SD)			HMG (M ± SD)			SMG (M ± SD)		
		*n* = 28			*n* = 30			*n* = 27		
Age		20.71 (1.16)			20.93 (0.73)			20.41 (1.37)		
Sex		Male = 10, female = 18			Male = 9, female = 21			Male = 8, female = 19		
Measure		Baseline	Post-induction	post-intervention	Baseline	Post-induction	post-intervention	Baseline	Post-induction	post-intervention
PANAS	PA	2.35 (0.64)	2.05 (0.63)	1.92 (0.65)	2.36 (0.65)	1.87 (0.62)	2.09 (0.75)	2.32 (0.71)	1.91 (0.56)	1.71 (0.69)
	NA	1.60 (0.51)	2.09 (0.73)	1.28 (0.40)	1.36 (0.38)	1.72 (0.51)	1.19 (0.28)	1.34 (0.30)	1.82 (0.57)	1.45 (0.41)
TMS		31.00 (5.29)		34.93 (4.46)	32.73 (6.05)		35.20 (5.29)	32.15 (3.63)		34.67 (4.04)

Note. PANAS: Positive and Negative Affect Schedule, PA: positive affect, NA: negative affect, TMS: Toronto Mindfulness Scale; CMG: calm music group, HMG: happy music group, SMG: sad music group; M: mean, SD: standard deviation.

**Table 2 ijerph-18-07063-t002:** Descriptive statistics of emotional Stroop task.

Variable	CMG (M ± SD)			HMG (M ± SD)			SMG (M ± SD)		
	Baseline	Post-Indunction	Post-Intervention	Baseline	Post-Indunction	Post-Intervention	Baseline	Post-Indunction	Post-Intervention
ACC	0.92 (0.02)	0.89 (0.02)	0.93 (0.01)	0.92 (0.01)	0.90 (0.02)	0.94 (0.01)	0.93 (0.02)	0.90 (0.02)	0.94 (0.01)
RTS	633.16 (18.66)	648.72 (18.26)	615.21 (17.32	669.28 (18.03)	675.73 (17.64)	637.06 (16.73)	668.35 (19.00)	678.14 (18.59)	624.80 (17.64)

Note. ACC: accuracy; RTs: reaction times; CMG: calm music group, HMG: happy music group, SMG: sad music group; M: mean, SD: standard deviation.

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
