# Peer review of "Regulation of Mindfulness-Based Music Listening on Negative Emotions Related to COVID-19: An ERP Study"

_ijerph, 2021, doi:10.3390/ijerph18137063_

Round 1
Reviewer 1 Report
The manuscript describes an interesting ERP study on the impact of different kinds of music (calm, happy, and sad) on the neural processing of a face-word emotional Stroop task. The authors investigated behavioral parameters like positive and negative mood and mindfulness as well as accuracy and reaction times during the Stroop task together with the neural parameters N2, N3, P3 and LPC before the induction of a negative mood by a video regarding Covid-19 pandemic (baseline), after the induction (post-induction) and after the presentation of a musical piece (post-intervention).
The introduction is well written and comprehensible. I would, however, suggest better elaborating the differences between the P3 and LPC component in order to introduce these two ERP components in a more detailed way. In line 170 and 171 the two components are even put on the same level. Please also indicate whether the LPC corresponds to the LPP, which is often found in emotional processing contexts.
The methods section is clearly written. In the EEG recording and analyses section please integrate a figure including all measured electrodes. Furthermore, in this section, it was not clear to me why authors only analyze 3 frontal electrodes (FC3, FCz, and FC4) even though they measured 32 electrodes. As the analyzed ERP components often show a posterior distribution in dependence of the functional interpretation (especially the P3 and LPC but also the N2 is it reflects more visually driven processing) I would suggest to run the same ANOVAs but including more electrodes all over the scalp. Maybe the authors might consider creating regions of interest (ROIs) in frontal, central and posterior regions and to introduce ROI as a factor in the ANOVA. This would provide a more details picture about the several processes involved during the Stroop task.
The formatting of Table 1 and Table 2 got out of place at some lines. Please reformat the tables by either using abbreviations for the variable names and putting the parenthesis in the same lines.
In Figure 4 the names post-induction and post-intervention in the color-coding legend on the right is missing. Please add these names.
In the legend of Figure 6, the name N450 is mentioned. I would suggest to use the same names as in the figure itself and as in the manuscript, so please change N450 into N3.
For completeness reasons, in the results section, please indicate additionally to the p-values also the df and t-values for posthoc t-tests.
The discussion section provides an overview over the main results. However, it necessitates some clarifications:
- In lines 564-567 the authors mention that a greater LPC was found for incongruent emotional stimuli of post-induction and post-intervention for the SMG. I cannot find any mentioning of this interaction between stimulus, measure and group in the results section. If the result section is correct there was no such an interaction. Thus, such an interpretation (see also lines 656-659) cannot be reported. Please change accordingly.
- The discussion section will also change a little but when authors integrate the results from the new ANOVAs also including central and posterior electrodes. Please integrate also these results in the discussion section.
- With respect to the N2 component, authors nicely report the three subcomponents of the N2. I, however, missed a statement which subcomponent was found in the present study in order to provide a more clear interpretation in regard.
- I missed a more detailed interpretation of the stimulus effects, especially with respect to the N2, N3, and LPC. Please integrate also these findings from our study.
Minor changes:
In some citations in the text, space characters are missing between citations or before citations in parenthesis (e.g., lines 108, 130, 141, 170, 197 etc.). Please check this throughout the whole manuscript.
In line 107 the word also should be removed once in: “… the Stroop task can also be used to investigate…”
Author Response
Dear referees:
Thank you for your insightful feedback on our manuscript. We have revised the manuscript in response to your comments, as outlined in the detailed, point-by-point response below, including revising the logic of the introduction section and citing additional literature, as raised by the reviewers. We indicated the main changes made in the revised paper using red font. We re-analyzed the ERP components of the five regions related to emotional regulation (frontal, frontal-central, central, central-parietal, and parietal), as suggested by the reviewers.
In addition, we shortened the manuscript as much as possible without greatly detracting from the content; we hope this revision improves readability. The manuscript was reviewed by a native English professional academic editor before resubmission. We hope the manuscript now meets the requirements for publication in your journal. We are available to address any further comments and suggestions.
Sincerely,
Maoping Zheng
2021.6.11
Comments and Suggestions for Authors
The manuscript describes an interesting ERP study on the impact of different kinds of music (calm, happy, and sad) on the neural processing of a face-word emotional Stroop task. The authors investigated behavioral parameters like positive and negative mood and mindfulness as well as accuracy and reaction times during the Stroop task together with the neural parameters N2, N3, P3 and LPC before the induction of a negative mood by a video regarding Covid-19 pandemic (baseline), after the induction (post-induction) and after the presentation of a musical piece (post-intervention).
The introduction is well written and comprehensible. I would, however, suggest better elaborating the differences between the P3 and LPC component in order to introduce these two ERP components in a more detailed way. In line 170 and 171 the two components are even put on the same level. Please also indicate whether the LPC corresponds to the LPP, which is often found in emotional processing contexts.
Response: Thank you for your suggestions. We revised the Introduction accordingly (see P4 L160-188). In terms of ERP components, late positive component (LPC) and late positive potential (LPP) are different expressions of the same EEG component. LPC/LPP is a positive potential occurring approximately 600-1000 ms after stimuli appearance (Liu et al., 2019a; Liu et al., 2019b; Deng et al, 2015).
The references are as follows:
Liu, Y., Quan, H., Song, S., Zhang, X., & Chen, H. (2019). Decreased Conflict Control in Overweight Chinese Females: Behavioral and Event-Related Potentials Evidence. Nutrients, 11(7), 1450. doi:10.3390/nu11071450
Deng, X., Ding, X., & Sang, B. (2015). The Implications of the Spatial-Temporal Shifting Patterns of Late Positive Potential (LPP) in the Study of Emotion Regulation Development. Journal of Psychological Science, 38(04), 853-860. doi:10.16719/j.cnki.1671-6981.2015.04.012
The methods section is clearly written. In the EEG recording and analyses section please integrate a figure including all measured electrodes. Furthermore, in this section, it was not clear to me why authors only analyze 3 frontal electrodes (FC3, FCz, and FC4) even though they measured 32 electrodes. As the analyzed ERP components often show a posterior distribution in dependence of the functional interpretation (especially the P3 and LPC but also the N2 is it reflects more visually driven processing) I would suggest to run the same ANOVAs but including more electrodes all over the scalp. Maybe the authors might consider creating regions of interest (ROIs) in frontal, central and posterior regions and to introduce ROI as a factor in the ANOVA. This would provide a more details picture about the several processes involved during the Stroop task.
Response: We re-analyzed the ERP components of five regions [frontal (F3, Fz, F4), frontal-central (FC3, FCz, FC4), central (C3, Cz, C4), central-parietal (CP3, CPz, CP4), and parietal (P3, Pz, P4)] as suggested, and revised and added relevant information to the Results (see P11 L470-482; P12 L483-515; P13 L516-553).
The formatting of Table 1 and Table 2 got out of place at some lines. Please reformat the tables by either using abbreviations for the variable names and putting the parenthesis in the same lines.
In Figure 4 the names post-induction and post-intervention in the color-coding legend on the right is missing. Please add these names.
Response: Per your suggestion, we revised the corresponding information in all figures (see P7 L337-339; P9 L409-413; P10 L431-433; P11 L461-163; L464-466; P12 L489-492) and tables (see P9 L408; P10 L447-448 ).
In the legend of Figure 6, the name N450 is mentioned. I would suggest to use the same names as in the figure itself and as in the manuscript, so please change N450 into N3.
Response: Thank you for the suggestion; we revised the legend of Figure 6 accordingly (see P12 L490).
For completeness reasons, in the results section, please indicate additionally to the p-values also the df and t-values for posthoc t-tests.
Response: Thank you for the suggestion; we added the corresponding information to the Results (see P11 L470-482; P12 L483-515; P13 L516-553).
The discussion section provides an overview over the main results. However, it necessitates some clarifications:
- In lines 564-567 the authors mention that a greater LPC was found for incongruent emotional stimuli of post-induction and post-intervention for the SMG. I cannot find any mentioning of this interaction between stimulus, measure and group in the results section. If the result section is correct there was no such an interaction. Thus, such an interpretation (see also lines 656-659) cannot be reported. Please change accordingly.
- The discussion section will also change a little but when authors integrate the results from the new ANOVAs also including central and posterior electrodes. Please integrate also these results in the discussion section.
- With respect to the N2 component, authors nicely report the three subcomponents of the N2. I, however, missed a statement which subcomponent was found in the present study in order to provide a more clear interpretation in regard.
- I missed a more detailed interpretation of the stimulus effects, especially with respect to the N2, N3, and LPC. Please integrate also these findings from our study.
Response: Thank you for the suggestion; we added the corresponding information to the Discussion (see P14 L579-619; P15 L620-637).
Minor changes:
In some citations in the text, space characters are missing between citations or before citations in parenthesis (e.g., lines 108, 130, 141, 170, 197 etc.). Please check this throughout the whole manuscript.
In line 107 the word also should be removed once in: “… the Stroop task can also be used to investigate…”
Response: Thank you for the suggestion, we rechecked citations in parentheses and removed redundant information.
Reviewer 2 Report
This study is expertly designed and done. I personally enjoyed reading this manuscript. In addition, this is a timely study given the still on-going pandemic. I have three suggestions.
1. The study made a strong claim on its implication on Covid-19 based on two facts: (a) the participants watched a video of Covid-19 in the post-induction Stroop task, and (b) all the participants were routinely exposed to stress related to Covid-19 in the past year. However, it is unclear whether the current findings are related to the negative mood specific to Covid-19 or a negative mood induced by an unpleasant experience in general. Thus, the authors should acknowledge this possibility in the discussion. In addition, since people are under the stress caused by Covid-19 worldwide, fact (b) described in the manuscript is not specific to the participants of this study at all. Thus, the authors should tone down the strong claims made on Covid-19 throughout this manuscript.
2. Figure 1 is confusing. As far as I see, the current study imposed Chinese characters on emotional faces. If this is true, Figure 1 should reflect what the participants were shown exactly. Please use Chinese characters in Figure 1.
3. The citations supporting the Stroop task are inadequate. Please cite the following articles that used the same experimental paradigm.
Yan, Y., Li, Y., Lou, X., Li, S., Yao, Y., Gong, D., Ma, W., & Yan, G. (In press). The Influence of action video gaming experience on the perception of emotional faces and emotional word meaning. Neural Plasticity.
R. Zhu, H. J. Zhang, T. T. Wu, W. B. Luo, and Y. J. Luo, “Emotional conflict occurs at an early stage: evidence from the emotional face–word Stroop task,” Neuroscience Letters, vol. 478, no. 1, pp. 1–4, 2010.
Author Response
Dear referees:
Thank you for your insightful feedback on our manuscript. We have revised the manuscript in response to your comments, as outlined in the detailed, point-by-point response below, including revising the logic of the introduction section and citing additional literature, as raised by the reviewers. We indicated the main changes made in the revised paper using red font. We re-analyzed the ERP components of the five regions related to emotional regulation (frontal, frontal-central, central, central-parietal, and parietal), as suggested by the reviewers.
In addition, we shortened the manuscript as much as possible without greatly detracting from the content; we hope this revision improves readability. The manuscript was reviewed by a native English professional academic editor before resubmission. We hope the manuscript now meets the requirements for publication in your journal. We are available to address any further comments and suggestions.
Sincerely,
Maoping Zheng
2021.6.11
Comments and Suggestions for Authors
This study is expertly designed and done. I personally enjoyed reading this manuscript. In addition, this is a timely study given the still on-going pandemic. I have three suggestions.
- The study made a strong claim on its implication on Covid-19 based on two facts: (a) the participants watched a video of Covid-19 in the post-induction Stroop task, and (b) all the participants were routinely exposed to stress related to Covid-19 in the past year. However, it is unclear whether the current findings are related to the negative mood specific to Covid-19 or a negative mood induced by an unpleasant experience in general. Thus, the authors should acknowledge this possibility in the discussion. In addition, since people are under the stress caused by Covid-19 worldwide, fact (b) described in the manuscript is not specific to the participants of this study at all. Thus, the authors should tone down the strong claims made on Covid-19 throughout this manuscript.
Response: Following your suggestion, we revised the information related to COVID-19 in the Introduction (see P1 L30-42; P2 L43-56).
- Figure 1 is confusing. As far as I see, the current study imposed Chinese characters on emotional faces. If this is true, Figure 1 should reflect what the participants were shown exactly. Please use Chinese characters in Figure 1.
Response: Following your suggestion, we revised Figure 1 in the face-word Stroop task and cited the below article (P7 L337-339 )
Yan, W., & Luo, Y. (2005). Standardization and Assessment of College Students' Facial Expression of Emotion. Chinese Journal of Clinical Psychology, 13(4), 396-398.
Zhu, X. R., Zhang, H. J., Wu, T. T., Luo, W. B., & Luo, Y. J. (2010). Emotional conflict occurs at an early stage: evidence from the emotional face-word Stroop task. Neuroscience Letters, 478(1), 1-4.
- The citations supporting the Stroop task are inadequate. Please cite the following articles that used the same experimental paradigm.
Yan, Y., Li, Y., Lou, X., Li, S., Yao, Y., Gong, D., Ma, W., & Yan, G. (In press). The Influence of action video gaming experience on the perception of emotional faces and emotional word meaning. Neural Plasticity.
- Zhu, H. J. Zhang, T. T. Wu, W. B. Luo, and Y. J. Luo, “Emotional conflict occurs at an early stage: evidence from the emotional face–word Stroop task,” Neuroscience Letters, vol. 478, no. 1, pp. 1–4, 2010.
Response: We cited the above article (see P7 L307) in the revised manuscript, based on your suggestion.
Reviewer 3 Report
The topic of the present paper is interesting. However, it deals with numerous concepts and standpoints which makes it difficult to draw proper conclusions. In addition, several shortcomings prevent me to recommend it for publication. Some examples are explained below:
The goal of the paper is not clear. Although in the title appears the word “Covid” and the introduction begins citing some works related to the disease, this question is quickly disregarded in the ongoing text. Indeed, no explanation about the particularities of Covid concerning emotions is included, such as the influence of having suffered the disease or having relatives with this problem, the fact of working in health context, the influence of lockdown, etc. I think it is essential to control all this factors in order to state that authors deal with the regulation of negative emotions related to Covid-19. Otherwise, without these considerations, the work is rather a study on negative emotions “in general” by using a video to modulate the affective state. Nonetheless, if the study was a fundamental study on emotions, the novelty and how the knowledge in the field will be increased by these findings is not evident.
The introduction is too long and too many references are cited, e.g., in lines 61 – 64, without indicating why these references are relevant in the specific context of the study. The citations are too general or ambiguous (e.g., “Additionally, previous studies have shown a relationship between mindfulness meditation and specific music activities”). It would be better to explain in detail some of the crucial results in the literature which lead to the hypotheses of the study, instead of showing a list of references. It is also important to avoid redundant information, such as the results presented in paragraph 66 – 88. Furthermore, from my point of view, several sentences are not relevant for the topic, e.g., the questions raised in lines 125 – 132 will not be discussed in the paper, the mention to the object processing, the link between the text 109 – 120 and Covid and negative emotions is not clear, etc.
In contrast, the choice of using music is not justified in the context of Covid. Do the authors think that it is better than other kind of regulation method? Actually, in my opinion, emotional state, emotional reaction and emotion regulation are some concepts which are hardly distinguished in the manuscript, making it difficult to follow the rationale of the study.
The sentence in lines 149 – 150 is not clear to me.
Concerning the hypotheses. How the authors could verify “No significant difference will be found between baseline and post-intervention in within-subject’s conflict control performance in the face-word Stroop task”. To my knowledge, ANOVA or t-tests are only adapted to claim significant differences and it is not possible to claim the opposite with this approach. Why not to mention HMG in the hypotheses? How the authors can operationalize the last hypothesis to verify the higher-order cognitive processes?
In regard to the methods, given the utilisation of unstandardized material, the replication would be very difficult. For instance, for the video, more details are needed (or supplementary material uploaded). Did the video show hospitals, ill people? Was there some music, speech, silence?
In line 326, what “during the go trials” means?
How can the authors take into account a possible training effect in the responses in the Stroop task?
It is not clear the method to reject artefacts for EEG, since in the manuscript we find: “if they included electro-oculogram (EOG) artefacts (ocular movements and eye blinks)”.
Was P3 measured in frontal region? Is this the best region to measured it?
For the ANOVA, I consider that, instead of using the factor “stimulus” with 4 levels, it would be more suitable to use 2 factors: “congruency” (with the levels “congruent” and “incongruent”) and “valence” (with the levels “sad” and “happy”). This would permit to discuss the main effects of valence or congruency in a proper way.
When was the PANAS filled? Before or after the Stroop task? Which is the importance of this choice? In addition to the assessment of the emotional state, it would have been expected a validation of the emotional content of the video or music by the participants.
Please verify the format of tables to make them readable. The legend of the figure 4 is missing. Where can we see the results on TMS mentioned in the caption of figure 2?
Is it correct to express p=0.000? Usually, we find p<.001 instead.
Overall, the discussion should be widely improved:
to avoid repeat directly the results (e.g., second paragraph),
to show the actual findings (since it focuses on other studies, e.g., 579 – 582; 614 – 618),
to avoid overstatements (e.g., 592 – 593; 608 – 609)
and to link the results with emotion regulation (e.g., in 620 – 630).
As in the introduction, too many references are cited in the discussion.
Verify reference format.
Author Response
Dear referees:
Thank you for your insightful feedback on our manuscript. We have revised the manuscript in response to your comments, as outlined in the detailed, point-by-point response below, including revising the logic of the introduction section and citing additional literature, as raised by the reviewers. We indicated the main changes made in the revised paper using red font. We re-analyzed the ERP components of the five regions related to emotional regulation (frontal, frontal-central, central, central-parietal, and parietal), as suggested by the reviewers.
In addition, we shortened the manuscript as much as possible without greatly detracting from the content; we hope this revision improves readability. The manuscript was reviewed by a native English professional academic editor before resubmission. We hope the manuscript now meets the requirements for publication in your journal. We are available to address any further comments and suggestions.
Sincerely,
Maoping Zheng
2021.6.11
Comments and Suggestions for Authors
The topic of the present paper is interesting. However, it deals with numerous concepts and standpoints which makes it difficult to draw proper conclusions. In addition, several shortcomings prevent me to recommend it for publication. Some examples are explained below:
The goal of the paper is not clear. Although in the title appears the word “Covid” and the introduction begins citing some works related to the disease, this question is quickly disregarded in the ongoing text. Indeed, no explanation about the particularities of Covid concerning emotions is included, such as the influence of having suffered the disease or having relatives with this problem, the fact of working in health context, the influence of lockdown, etc. I think it is essential to control all this factors in order to state that authors deal with the regulation of negative emotions related to Covid-19. Otherwise, without these considerations, the work is rather a study on negative emotions “in general” by using a video to modulate the affective state. Nonetheless, if the study was a fundamental study on emotions, the novelty and how the knowledge in the field will be increased by these findings is not evident.
Response: Thank you for the suggestion; we revised the corresponding information in the Introduction (see P1 L30-42; P2 L43-88).
The introduction is too long and too many references are cited, e.g., in lines 61 – 64, without indicating why these references are relevant in the specific context of the study. The citations are too general or ambiguous (e.g., “Additionally, previous studies have shown a relationship between mindfulness meditation and specific music activities”). It would be better to explain in detail some of the crucial results in the literature which lead to the hypotheses of the study, instead of showing a list of references. It is also important to avoid redundant information, such as the results presented in paragraph 66 – 88. Furthermore, from my point of view, several sentences are not relevant for the topic, e.g., the questions raised in lines 125 – 132 will not be discussed in the paper, the mention to the object processing, the link between the text 109 – 120 and Covid and negative emotions is not clear, etc.
Response: Many thanks for your suggestion, we have shortened the manuscript as much as possible without greatly affecting the content. We also have revised the corresponding information to the introduction and removed redundant information.
In contrast, the choice of using music is not justified in the context of Covid. Do the authors think that it is better than other kind of regulation method? Actually, in my opinion, emotional state, emotional reaction and emotion regulation are some concepts which are hardly distinguished in the manuscript, making it difficult to follow the rationale of the study.
Response: Previous studies have shown that effective music listening can effectively improve negative emotions. Based on the behavioral results of existing studies (Carlson et al., 2015; Groarke & Hogan, 2019), we aimed to explore the cognitive neural correlation of music with different emotional valence levels on negative emotion regulation using Stroop paradigm. Based on your suggestions, we added explanations and discussions about emotional state, response, and regulation to the Introduction and Discussion.
Carlson, E., Saarikallio, S., Toiviainen, P., Bogert, B., Kliuchko, M., & Brattico, E. (2015). Maladaptive and adaptive emotion regulation through music: a behavioral and neuroimaging study of males and females. Front Hum Neurosci, 9, 466. doi:10.3389/fnhum.2015.00466
Groarke, J. M., & Hogan, M. J. (2019). Listening to self-chosen music regulates induced negative affect for both younger and older adults. PLoS One, 14(6), e0218017. doi:10.1371/journal.pone.0218017
The sentence in lines 149 – 150 is not clear to me.
Response: We have removed the corresponding information (see P3 L152 ).
Concerning the hypotheses. How the authors could verify “No significant difference will be found between baseline and post-intervention in within-subject’s conflict control performance in the face-word Stroop task”. To my knowledge, ANOVA or t-tests are only adapted to claim significant differences and it is not possible to claim the opposite with this approach. Why not to mention HMG in the hypotheses? How the authors can operationalize the last hypothesis to verify the higher-order cognitive processes?
Response: Following your advice, we revised the hypotheses (see P5 L218-233). Previous studies (Hajcak & Nieuwenhuis, 2006; Huang & Luo, 2006) have shown that LPC is an EEG component that reflects advanced cognitive processing (see P4 L184-189 ).
In regard to the methods, given the utilisation of unstandardized material, the replication would be very difficult. For instance, for the video, more details are needed (or supplementary material uploaded). Did the video show hospitals, ill people? Was there some music, speech, silence?
Response: Following your advice, we uploaded supplementary material (a 4 minute video excerpt related to COVID-19). The video is compiled based on excerpts from the COVID-19 epidemic in China in 2020 (see P5 L246-252; P6 L157-258).
In line 326, what “during the go trials” means?
Response: We removed this information (see P7 L329).
How can the authors take into account a possible training effect in the responses in the Stroop task?
Response: In order to eliminate the training effect of the Stroop task as much as possible, we performed a completely random presentation within and between subjects in different Stroop task phases, and added an intervention task of facial emotion intensity.
It is not clear the method to reject artefacts for EEG, since in the manuscript we find: “if they included electro-oculogram (EOG) artefacts (ocular movements and eye blinks)”.
Response: We revised the corresponding information (see P8 L382-387).
Was P3 measured in frontal region? Is this the best region to measured it?
Response: Following your suggestion, we re-analyzed the ERP components of five regions [frontal (F3, Fz, F4), frontal-central (FC3, FCz, FC4), central (C3, Cz, C4), central-parietal (CP3, CPz, CP4), and parietal (P3, Pz, P4)] and revised and added corresponding information to the Results (see P9 L391-399).
For the ANOVA, I consider that, instead of using the factor “stimulus” with 4 levels, it would be more suitable to use 2 factors: “congruency” (with the levels “congruent” and “incongruent”) and “valence” (with the levels “sad” and “happy”). This would permit to discuss the main effects of valence or congruency in a proper way.
Response: We re-analyzed the Stroop task data and revised and added corresponding information to the Results (see P11 L470-482; P12 L483-515; P13 L516-553).
When was the PANAS filled? Before or after the Stroop task? Which is the importance of this choice? In addition to the assessment of the emotional state, it would have been expected a validation of the emotional content of the video or music by the participants.
Response: We revised corresponding information related to the procedure (see P8 L343-354).
Please verify the format of tables to make them readable. The legend of the figure 4 is missing. Where can we see the results on TMS mentioned in the caption of figure 2?
Is it correct to express p=0.000? Usually, we find p<.001 instead.
Response: We revised the corresponding information (see P9 L409-413; P11 L461-463).
Overall, the discussion should be widely improved:
to avoid repeat directly the results (e.g., second paragraph),
to show the actual findings (since it focuses on other studies, e.g., 579 – 582; 614 – 618),
to avoid overstatements (e.g., 592 – 593; 608 – 609)
and to link the results with emotion regulation (e.g., in 620 – 630).
As in the introduction, too many references are cited in the discussion.
Verify reference format.
Response: Following your suggestion, we revised corresponding information (see P13 L554-565; P14 L569-570, L579-595, L650-613, L618-619; P15 L620-624, L630-637, L650-658).
Round 2
Reviewer 1 Report
The authors addressed some but not all of my concerns of the first review round.
The introduction section was extended and includes more relevant specifications regarding the several subthemes. However, the section regarding the explanation of P3 and LPC is rather confusing. The description of the LPC is now better comprehensible and indicates in a clearer way the underlying emotional processes. With respect to the P3 authors describe similar processes (e.g., of cognitive evaluation of the stimuli’s meaning) as for the LPC. It is not clear whether P3 is indeed found also in studies using emotional stimuli or whether the P3 represents a more domain-general attentional marker, independent from emotional processing and what the exact differences between P3 and LPC are (with respect to timing, topography, experimental sensitivity, and underlying processes). I think the authors should elaborate this issue in a more extensive way, maybe integrating more details about the studies reported in order to clearly differentiate the mechanisms underlying these two ERP components.
Furthermore, could you please indicate in the introduction that the LPC and LPP represent the same ERP component. This is important for a clear comprehension across several studies using these different nomenclatures for the same ERP component.
I appreciate that authors now reanalyzed the data including regions of interest (ROIs) such as frontal, fronto-central, central, central-parietal and parietal sites. In line 389 onwards, authors also describe that each ROI consists of 3 different electrodes. Such an approach is fine. However, when reporting the results of the simple effects analyses following an interaction including the factor electrode site, authors report effects at a single electrode level. In my understanding, authors averaged the 3 electrodes of each ROI for conducting the ANOVA. As a consequence, the simple effect analyses should be performed on these ROIs (thus, with averaged electrodes) and not on selective single electrodes.
Furthermore, in the first review round I suggested to include a figure including the 32 electrodes measured. Unfortunately, the authors did not follow my suggestion. If author do not want to do such an additional figure, I however, suggest adding at least the names of the 32 electrodes recorded in line 370 in order to provide the reader with the exact localization of the electrodes selected.
Furthermore, in the first review round I asked the authors to include df- and t-values also for the posthoc t-tests performed (thus, for the simple effect analyses), not only for the ANOVAs. As this was not present in the revised version, I kindly ask again to include these data for completeness reasons, also to potentially evaluate whether there were missing cases or not.
The discussion section is difficult to evaluate in its entirety as the posthoc results are not reported for ROIs but erroneously for selective single electrodes. However, I noted that the interpretation of the LPC component was definitely improved in the revised version and the explanation with respect to increased arousal for emotional stimuli seems plausible. An interpretation of the N2 reflecting an attentional bias also seems rather traceable. However, the interpretation of the N3 as “a variable for studying emotional processing to visual stimuli” is too vague. Please add more literature to clearly differentiate the processes underlying the N3 from those underlying the N2. Similarly, a clear differentiation between the processes underlying the P3 and LPC is missing. The authors even describe the same processes for both components (e.g., cognitive evaluation of the meaning of stimuli; lines 613 and 624). Which kind of processes does the P3 exactly represents in the study context? And how do they differentiate from the LPC. I think, in this regard, more elaboration including more studies and describing their materials, designs etc. in a more detailed fashion is needed.
Author Response
Dear reviewer 1:
Thank you for your insightful feedback on our manuscript. We have revised and added information concerning the ERPs (N3 and P3), as suggested in your comments. We have indicated the main changes made in the revised paper using a red font color. This revised manuscript has gone through professional English editing. We hope the revised manuscript meets the requirements for publication in your journal. We are available to address any further comments and suggestions.
Sincerely,
Maoping Zheng
2021.6.21

Reviewer 3 Report
I acknowledge the effort made by the authors to improve the manuscript. The method is more detailed and the results section has been widely improved, although I do not understand which is the factor "trials".
However, I have the impression that my previous comments about the introduction, and consequently about the discussion, have not been properly addressed (or refuted) and the authors have rather replaced some paragraphs. Therefore I invite the authors to reconsider my suggestions (see below) concerning the goal of the study, the link to Covid and the quantity of references.
The goal of the paper is not clear. Although in the title appears the word “Covid” and the introduction begins citing some works related to the disease, this question is quickly disregarded in the ongoing text. Indeed, no explanation about the particularities of Covid concerning emotions is included, such as the influence of having suffered the disease or having relatives with this problem, the fact of working in health context, the influence of lockdown, etc. I think it is essential to control all these factors in order to claim that authors deal with the regulation of negative emotions related to Covid-19. Otherwise, without these considerations, the work is rather a study on negative emotions “in general” by using a video to modulate the affective state. Nonetheless, if the study was a fundamental study on emotions, the novelty and how the knowledge in the field will be increased by these findings is not evident.
In contrast, the choice of using music is not justified in the context of Covid. Do the authors think that it is better than other kind of regulation method?"
Please, verify some typos, e.g., line 629.
Author Response
Dear reviewer 3:
Thank you for your insightful feedback on our manuscript. We have revised and added the information regarding the goal of the study, especially its relation to Covid-19, and the number of references has increased, as raised in your comments. We indicated the main changes made in the revised paper using a red font color. This revised manuscript has gone through professional English editing. We hope the revised manuscript meets the requirements for publication in your journal. We are available to address any further comments and suggestions.
Sincerely,
Maoping Zheng
2021.6.21
